# State of the Art of the Molecular Biology of the Interaction between Cocoa and Witches’ Broom Disease: A Systematic Review

**DOI:** 10.3390/ijms24065684

**Published:** 2023-03-16

**Authors:** Ariana Silva Santos, Irma Yuliana Mora-Ocampo, Diogo Pereira Silva de Novais, Eric Roberto Guimarães Rocha Aguiar, Carlos Priminho Pirovani

**Affiliations:** 1Centro de Biotecnologia e Genética (CBG), Departamento de Ciências Biológicas (DCB), Universidade Estadual de Santa Cruz (UESC), Ilhéus 45662-900, BA, Brazil; 2Instituto Federal de Educação Ciência e Tecnologia da Bahia (IFBA), Porto Seguro 45810-000, BA, Brazil

**Keywords:** *Theobroma*, witches’ broom, molecular interaction, plant-pathogen interaction

## Abstract

Significant scientific advances to elucidate the *Moniliophthora perniciosa* pathosystem have been achieved in recent years, but the molecular biology of this pathogen-host interaction is still a field with many unanswered questions. In order to present insights at the molecular level, we present the first systematic review on the theme. All told, 1118 studies were extracted from public databases. Of these, 109 were eligible for the review, based on the inclusion and exclusion criteria. The results indicated that understanding the transition from the biotrophic-necrotrophic phase of the fungus is crucial for control of the disease. Proteins with strong biotechnological potential or that can be targets for pathosystem intervention were identified, but studies regarding possible applications are still limited. The studies identified revealed important genes in the *M. perniciosa*-host interaction and efficient molecular markers in the search for genetic variability and sources of resistance, with *Theobroma cacao* being the most common host. An arsenal of effectors already identified and not explored in the pathosystem were highlighted. This systematic review contributes to the understanding of the pathosystem at the molecular level, offering new insights and proposing different paths for the development of new strategies to control witches’ broom disease.

## 1. Introduction

“All you need is love. But a little chocolate now and then doesn’t hurt”, is a famous saying by the American cartoonist Charles Schulz. In the field of science, the production of chocolate and cupulate, the main products made from plant species of the genus *Theobroma*, are threatened by witches’ broom disease (WBD) caused by *Moniliophthora perniciosa*. Despite this scenario, the Zion Market Research Report [1] indicates that the international chocolate trade was worth of US$ 136.01 billion in 2020 and is forecast to generate revenue of around US$ 192.12 billion in 2028.

Despite this optimistic international outlook, the devastating fungus, *M. perniciosa* reduced the production of the main raw material, the seeds, by approximately 190,000 tons between 2017 and 2020 [2]. In addition, ICCO [2] warns that the fall in world cocoa production resulting from less favorable weather conditions and increased presence of diseases that affect cocoa trees, such as WBD, will cause a 6.8% decline in real revenue in 2022 (after considering high inflation).

WBD was reported for the first time in Suriname in 1985. Soon thereafter, its presence was also reported in the Amazon region. However, in Brazil, more precisely in the south of the state of Bahia, it was only described in 1989 [3,4]. The geographic distribution of the disease is shaped by the physiology of the causal agent under different climate conditions [5].

WBD is a tropical disease resulting from the interaction between evergreen trees and the fungal pathogen *M. perniciosa*. The fungus is hemibiotrophic, with a very peculiar life cycle, because the majority of plant diseases caused by this class of fungi have two infection phases: (i) asymptomatic biotrophic phase (transient); and (ii) necrotrophic phase. This does not apply to *M. perniciosa*. The biotrophic phase is symptomatic and long-lasting, while less abundant hyphae remain in the apoplast, appropriating the nutrients available there, manipulating the host’s metabolism to generate greater availability of nutrients. This phase lasts an average of 60 days in the living tissues of the hosts. Then, at around 90 days of the molecular battle, the infected tissue becomes necrotic. The fungus undergoes dikaryotization and produces clamp connections, characterizing the second phase of the life cycle: the necrotrophic phase. At this stage, the fungus invades the intracellular space and undergoes primary homothallism as part of its reproductive process [6,7,8,9,10,11].

After the elucidation of the life cycle of *M. perniciosa*, studies were performed to understand its action mechanism. The infection and development of the fungus were orchestrated in favorable environmental conditions and the nutritional status of the host, resulting in physiological, histological and metabolic changes [12,13,14,15,16].

The molecular action mechanisms of the fungus have only recently become known, when the genomic sequences of *M. perniciosa* [10] and its main host, *Theobroma cacao* [17,18], were published, along with availability of the dual transcriptome *M. perniciosa* and *Theobroma cacao* [11]. These developments have brought the possibility of more detailed molecular studies of these organisms and their mechanisms of action.

Since then, several other studies have been published analyzing how biotrophy is maintained for so long in *M. perniciosa*, what the molecular mechanisms involved in the necrotrophic phase are, and the participation of hosts in signaling the end of these fungal phases [19,20,21,22,23].

In summary, the genome and dual transcriptome of *M. perniciosa* have already been identified, as well as its mitochondrial genome, its genetic variability, and polymorphic regions of the chromosomes [10,11,24,25,26]. Biochemical modifications of hosts when infected by *M. perniciosa* have also been investigated [27]. Cultivation protocols have been established, such as the artificial cultivation in cookies that allow the production of basidiocarps [28], followed by molecular studies of the fungus’ necrotrophic phase based on this type of cultivation [23,29]. With technological advances, improvements in the cultivation and manipulation techniques of the fungus have been achieved [30,31]. Identification of fungal genes through EST libraries has also occurred [32]. The challenge of in vitro production of the fungus in the biotrophic phase has been overcome [9]. Fungal protein profiles have been traced and key proteins related to pathogenicity and necrosis have been identified [21,22,23,33,34].

However, the existing information of this biological puzzle has never been systematized to enable a better understanding of the molecular biology of the fungus, and many questions about the molecular mechanisms of control and change are unresolved at the experimental level [10]. Therefore, we present a systematic review of this information to provide answers and improve understanding of the molecular biology of *M. perniciosa* and its interaction with the main hosts.

The meticulous and accurate nature of systematic reviews sets them apart from other types of reviews. A systematic review is a type of study that synthesizes and critically evaluates data compiled from previously published scientific investigations of a particular topic, providing a higher level of what is considered good evidence [35,36,37]. The first field of study to use systematic reviews was medicine, starting in the 18th century, and since then systematic reviews have been published in virtually all academic areas, including the biological sciences. Systematic reviews on the behavior of pathogens that affect important plant crops [38,39], as well as on the mitigation of the effects of abiotic factors [40,41,42], have contributed to the elucidation of molecular mechanisms, action mechanisms and host responses.

The collection of knowledge about the genetics, structural genomics, molecular mechanisms of action of the fungus and host defense, together with analysis of strategies and tools used, can help to fill gaps in knowledge about the *M. perniciosa* pathosystem. This systematic review summarizes the relevant literature published in recent years on the molecular biology of WBD caused by *M. perniciosa* and its impacts on hosts.

## 2. Methods

The systematic review was performed using the R program, with the *Bibliometrix* package and the StArt software (State of the Art through Systematic Review), Beta version 3.0.3, developed at Federal University of São Carlos (UFSCar), available at http://lapes.dc.ufscar.br/tools/start_tool (accessed on 5 May 2022). This review followed the PRISMA guidelines (Preferred Reporting Items for Systematic Reviews and Meta-Analyses) [43] and involved three fundamental steps: planning, execution, and summarization.

### 2.1. Planning

At this stage, a defined protocol was elaborated and discussed in a group. The protocol presented the basic information to guide this review, such as article title, authors, objective, keywords, search strings, research questions, research sources, inclusion/exclusion criteria, database selection and definition of the type of study (https://github.com/ArianaSantos/Santos-et-al.2022_systematic-review.git, accessed on 6 October 2022).

To achieve the objective of the systematic review, we formulated 14 questions, listed in Table 1.

These questions were based on the Population Intervention Comparison Results (PICOS) research strategy [44], as shown in Table 2. This strategy guided the elaboration of research questions and the specification to search for answers, avoiding biased responses [39,45].

### 2.2. Execution

In this step, we searched for studies using the string ((theobroma OR cacao OR cocoa OR moniliophthora OR crinipellis) AND (perniciosa OR broom)), in the databases Pubmed, Scopus, Scielo and Web Of Science. Boolean connectors “OR” and “AND” were used in the string to group synonymous keywords and main terms. The selected files were imported in BIBTEX and MEDLINE format into the R program.

After screening in R, the files with the same formats were exported to the Start program (v. Beta 3.0.3), where the automated selection was made based on the reading of titles and abstracts, using the inclusion and exclusion criteria established in the protocol as a reference (https://github.com/ArianaSantos/Santos-et-al.2022_systematic-review.git, accessed on 6 October 2022), to characterize the studies as accepted or rejected, and to exclude duplicates.

### 2.3. Summarization

Studies considered to be accepted for meeting one or more inclusion criteria were read completely to extract the data. This process consisted of extracting and summarizing the data that answered the review questions. During this phase, it was also possible to exclude some studies that, after a complete reading, met at least one exclusion criterion.

This step also included the elaboration of figures and tables in order to summarize the data found.

Data related to the production and dissemination of scientific knowledge were grouped by similarity with the Vosviewer software (version 1.16.17) [46]. The biological models were built based on those available on the Swissbiopics platform. (https://www.swissbiopics.org/, accessed on 15 September 2022).

The Protein–Protein Interaction Network (PPI) was obtained from the proteins identified in the selected studies. For this purpose: (i) searches were carried out with the names of the proteins in two databases—NCBI (https://www.ncbi.nlm.nih.gov/, accessed on 5 August 2022) and Uniprot (https://www.uniprot.org/, accessed on 5 August 2022)—to obtain the FASTA sequence; (ii) only proteins for the organisms *M. perniciosa* or *Moniliophthora* roreri were admitted in the searches, with the former being preferred; (iii) with the FASTA sequence of the proteins, BLAST was performed, using BLASTp (https://blast.ncbi.nlm.nih.gov/Blast.cgi?PAGE=Proteins, accessed on 5 August 2022), accepting only proteins with an identity percentage ≥90%; (iv) proteins that were not found through the search organisms terms or had <90% identity were not included in the PPI network.

With String v.11.0 (https://stringdbstatic.org, accessed on 19 August 2022) [47], another BLASTp was performed against the available *M. perniciosa* proteins. Proteins were analyzed individually with the following parameters: meaning of network edges: confidence; active interaction sources: textmining, experiments, databases, co-expression, neighborhood, gene fusion and co-occurrence; minimum required interaction score: high confidence (0.700); and max number of interactors shown for the 1st and 2nd shells: no more than 50 interactions. The file of each network was downloaded in TSV format and later the files were merged and analyzed using the Cytoscape software version 3.8.2 [48].

The centrality properties (betweenness and node degree) and modularity of the protein-protein interaction network were obtained through the igraph package of the R software (version 14.0.3) [49]. 

For the analysis of genetic ontology enrichment of each cluster in the network, the Analysis tool of the STRING database was used. Biological processes with a false discovery rate (FDR) value closest to 0 were assigned to the corresponding cluster. The FDR identifies how significant enrichment is. Corrected *p*-values for various tests within each category are shown using the Benjamini–Hochberg procedure (https://string-db.org, accessed on 19 August 2022).

To reduce any bias in the elaboration of the systematic review, we used the PRISMA checklist [43] (Appendix A, https://github.com/ArianaSantos/Santos-et-al.2022_systematic-review.git, accessed on 6 October 2022).

## 3. Results

### 3.1. Bibliometric Indicators

We initially identified a total of 1118 articles from searches in the databases Pubmed (175), Web of Science (463), Scopus (407) and Scielo (73). With the Bibliometrix package in the R software, 509 duplicated articles were excluded. After the first selection by the StArt software, another 500 articles were excluded because they were ineligible according to the inclusion criteria. The remaining 109 articles were included because they met at least one inclusion criterion (Figure 1).

Eligible studies were grouped into three categories of attributes, related to the reported element, study strategy and host (Figure 2A). To understand the biology of *M. perniciosa*, or its pathosystem, the main elements reported were genes in 50% of the studies, followed by morphological alterations and proteins, with 35% and 25%, respectively. From the elements reported in the 109 studies, the most discussed study strategies were gene expression, morphology, biochemistry, and proteomics, with 40%, 35%, 30% and 25% of the articles, respectively.

The studies that were published between 1959 and 2002 exclusively investigated morphology and/or biochemistry, but this field of study has still been present in studies over the past few years. Studies dating from 2003 have focused on the investigation of molecular markers to understand the biology of *M. perniciosa*. From 2007 onwards, functional genomics and proteomics strategies began to be used in order to elucidate the molecular mechanism of fungal action and interaction with the hosts.

The host species of greatest interest for the investigation of *M. perniciosa* biology are members of the genus *Theobroma* ssp. Indeed, *Theobroma cacao* L. or *Theobroma grandiflorum* alone accounted for ~90% of all articles that provided information on *M. perniciosa* biology (Figure 2A).

Approximately 80% of the studies included in this review were written by researchers from Brazil, which is the only country for which all study strategies among those categorized have been used to understand the mechanism of action of the fungus and its interaction with hosts. The countries to which most of these studies refer are located in the southern hemisphere in the tropical zone, followed by European countries such as France (with some partnership studies with Brazilian researchers) (Figure 2B).

Studies of *M. perniciosa* and its interaction with hosts date back to 1959, representing 0.9% of the publications. With the advent of Omics technologies, highlighted by the sequencing of the fungus’ genome, studies increased from 2007 onwards, accounting for 10% of the studies using different methodological strategies (Figure 2C). Since then, the number of studies per year has remained varied, aiming to elucidate the action and interaction of the *M. perniciosa* fungus with its hosts.

With the bibliometric data, we also systematized the journals that mostly published molecular studies of *M. perniciosa* and its hosts (Figure 3A). From the selected studies, the journals Physiology of Molecular Plant Pathology, Genetic and Molecular Research, PlosOne, Plant Pathology Biochemistry, and Plant Pathology contained the largest number of publications. The journals listed in Figure 3A have an impact factor >2.0, with the exception of Genetic and Molecular Research, which currently has an impact factor <1.5, but the studies included in this review that were published in this journal are from the years 2009 to 2014, during which the journal had a higher impact factor. Eligible publications for this work came from journals with peer review, high reliability indices and that fit in the scope of the study.

Figure 3B represents the research and teaching institutions that most have produced and collaborated with each other in studies of *M. perniciosa* and its hosts. The institutions involved in the 109 studies are grouped into nine clusters, represented by different colors. This grouping by colors was defined according to the collaboration patterns of these institutions, that is, institutions of the same group collaborating with each other and between similar institutions. As an example, we highlight the light blue color group, composed by the Comissão Executiva do Plano de Lavoura Cacaueira (CEPLAC), Universidade Federal de Viçosa (UFV), Universidade Estadual de Santa Cruz (UESC), Universidade Federal de Minas Gerais (UFMG), Universidade Estadual de Feira de Santana (UEFS) and Coodetec. These institutions, in addition to collaborating with each other in the development of studies, also have collaborated with all the other institutions present in other clusters, mainly with other institutions included in the pink group. The blue cluster has thick connectors, representing high frequency of collaboration. This is evident among the institutions of the pink, green and yellow groups, which are highly represented by the hubs of Universidade Estadual de Campinas (UNICAMP), Center for International Cooperation in Agronomic Research for Development (CIRAD), Empresa Brasileira de Agropecuária (Embrapa) and Mars Incorporated, respectively.

Within this collaboration network, the institutions UNICAMP, UESC, CEPLAC and CIRAD stand out mainly because their hub has a larger size, which means they are responsible for the largest number of studies included in this review.

In order to identify the profiles of the collaborations among the scientists who performed these studies, we created another network (Figure 3C). This network contains five groups, represented by different colors: green, yellow, purple, blue, and pink. The authors of the pink group have collaborated in research with all the other groups, but their publication index has the same pattern, since no hub stands out. The opposite occurs for some authors from the green, yellow, and purple groups, who have collaborated with each other and for which hubs stand out, denoting a high number of publications, such as C. Pirovani, F. Micheli and K. Gramacho in the green, yellow and purple groups, respectively.

### 3.2. The Biology of M. perniciosa and Its Hosts over Time

The development of the basidiomycete *M. perniciosa* was analyzed in 20 articles of this review, mainly in studies focusing on morphology (32%); biochemistry (30%) and physiology (6%) (Figure 2A).

The information that generally characterizes the fungal infection process in different hosts with compatible interaction over time is summarized in Figure 4. Differences regarding the development of infection in resistant genotypes will be discussed later.

The infection process that leads to the establishment of the WBD begins when the fungal spores come into contact with the meristematic regions of the hosts, mainly fruits, buds and floral pads. In the initial contact period (0–2 h), there are no visible symptoms [50]. In the period from 2–4 h, the spores adhere to the surface, and from 4–6 h they germinate and begin the process of penetration. Studies have shown that even without visible symptoms of infection, penetration is mechanical and enzymatic. The first occurs in multiple ways: through the base of the trichomes, the cuticle, natural epidermal openings, and even through the stoma1ta. The second occurs through pectinolytic enzymes, cellulases and xylanases, among others [50]. The period of penetration is characterized by the involvement of a substance deposited at the penetration site, representing the primary hyphae, with the presence of mucilage [50].

Once the fungus has entered the tissue, within two days after infection, primary hyphae are found in the cortex below the epidermal layer. This is characterized by the presence of macro and microscopic symptoms, visible in the hosts. WBD symptoms such as apical swelling, changes in bud morphology, hyphae growth towards vascular bundles and tissue hypertrophy become more pronounced in the period from 21 to 35 days after infection [50].

The infection advances further with time. At 45 days of infection, the phenotypic response of terminal green brooms is visible, as well as the formation of clamp connections and calcium oxalate crystals (Figure 4) [51]. At 60 days, after infection, necrosis begins at the tips of the leaves of the terminal broom and extends to affect other tissues. Dikaryotic hyphae are also formed and spread throughout the infected tissue. Afterwards, in the period of 60 to 90 days after infection, the green broom begins to dry out, becoming necrotic [50].

In the initial phase (0–2 h), the hyphae grow in the intercellular space of the hosts but do not yet present appressoria (Figure 4), which makes the fungus rely on the nutrients in the apoplast. In the final stage of infection (90 days), most hyphae become intracellular and necrotic symptoms are visible [50]. According to this morphological pattern, the studies correlate this condition with the greater morphological transition of the fungus from the biotrophic to the necrotrophic phase.

### 3.3. Interaction of Fungal Genes and Hosts

The studies that identified the genes of the fungus along with those of the hosts in the pathosystem used genomic analysis (5%) and gene expression techniques as strategies, which accounted for 40% of the studies.

In 27 studies, it was possible to count 264 genes identified and discussed in different hosts infected by the fungus. These genes were identified in the hosts *Theobroma cacao* (246 genes/groups), *Theobroma grandiflorum* (7 genes)*, Solanum lycopersicum* L. (10 genes) and *Nicotiana tabacum* (1 gene), where in most cases transcriptional activity was detected. For *M. perniciosa*, in 22 studies it was possible identify 373 genes of the life cycle, both in the biotrophic and necrotic phases, in different stages of development in the basidiospores and mycelia (Appendix A). However, the studies that identified genes of *M. perniciosa* were based on the infection process in the host *T. cacao*.

Genes reported in hosts infected with *M. perniciosa* had several functions. For example, genes with oxidoreductase function, such as ascorbate peroxidase (*apx*), glutathione peroxidase (*gpx*), and superoxide dismutase (*sod*); genes encoding resistance proteins, such as CC-NBS-LRR, NPR1/NIM1-interacting proteins, PRs-3, 4, 4b, 8 and 10; genes expressing transcription factors such as bZIP and WRKY; genes associated with cell wall modification; genes associated with vacuolar processing enzymes; genes encoding proteases such as legumains; genes encoding proteins with antimicrobial activity such as phylloplanin (TcPHYLL); genes encoding protease inhibitors, such as papain-like cysteine protease inhibitors and serine protease inhibitors; genes expressing chitinases, chaperones and ethylene response factors; as well as genes involved in the biosynthesis of metabolites and hormones (details and references in Appendix A).

In all studies, the genes identified showed differential expression throughout the infection, which varied according to the experimental design of the studies, including time frame, technical replicates, and biological replicates, among other factors (details and references in Appendix A).

*Moniliophthora perniciosa* activates a genetic arsenal to prevail in the molecular battle with the hosts. In the germination phase, the DNA polymerase and RNA polymerase genes were identified in some studies as upregulated compared to the investigated treatments [11,52]. Effector protein, chitin deacetylase 9, asparagine amidase a, oxidase, and plant pathogenesis related 1 protein also showed transcriptional activity in the fungal germination stage (Appendix A). In the biotrophic phase, several genes showed differential expression, based mainly on infection conditions such as time, strain and host. We highlight as functional examples of the genes identified: necrosis, with the *Mp-nep-1* and 2 genes [33]; GABA transport, with stomatin, cytochrome C oxidase, proteases and lipases; and pectin degradation, with alcohol oxidase, phosphoglycerate kinase, enolase, malate dehydrogenase and alternative oxidase [53]. In the necrotrophic phase of the disease, some of these genes’ expression are accentuated, such as *Mp-nep-1* and *2*, PRs, superoxide dismutases and catalase, while others are induced, such as genes related to cytochrome P450, gypsy-like retrotranspon, GMC oxidoreductase, serine threonine kinase, transposons, endoglucanases, and heat-shock protein HSS1 [10,53].

From the summary of these genes identified, it was possible to note that responses associated with molecular, biochemical, and physiological events occur in the process of WBD development, and these are mainly described in relation to hosts with compatible interaction, as shown in Figure 5. The studies that have contributed to the systematization of this information indicate that in the first (asymptomatic) stage of infection in the hosts, there is a significant increase in alkaloids, phenolic compounds and tannins in the cell, and sugars in the apoplast, a pattern that continues in the green broom phase, with an increase of ethylene, asparagine, sugars, malondialdehydes, theobromine and caffeine, as well as changes in fatty acid and amino acid profiles and plastid metabolism, advancing to programmed cell death (PCD) of infected tissues (Figure 5) [19,27].

In the initial phase of WBD, when phenotypic symptoms are not yet visible, proteins such as Pathogenesis-related protein 1 (MpPR-1) are induced in germinating spores [21]. More than 33 effector proteins are secreted by the fungus, aiming at successful spore germination [22]. In this phase, the action of the enzyme pectin methyl esterase (PME) is observed, which demethylates the pectin from the host cell wall [54]. On the other hand, methanol oxidase (MOX), induced by the fungus, metabolizes methanol, a byproduct of pectin degradation, to favor its entry and spore germination [54]. With spore germination, a cerato-platanin protein 5 (MpCP-5) is also induced to block the plant’s defense response by eliminating chitin fragments [55], as shown in Figure 5.

Overall, the responses were similar between different hosts, such as the high content of starch, glucose, fructose, and sucrose in the apoplast at the beginning of the infection, when there are still no visible symptoms [27]. The constitutive content of calcium oxalate crystals is higher in susceptible genotypes than in resistant genotypes [56]. However, this content increases throughout the biotrophic phase, and in resistant genotypes, calcium oxalate crystals it is diluted in the initial phase of infection. The analyses show regulation of the genes of antioxidant enzymes such as APX and GPX in response to the oxidative stress caused by the fungus, mainly through the action of MOX in this initial phase, since its activity leads to accumulation of reactive oxygen species (ROS), such as hydrogen peroxide (H_2_O_2_) [54,57].

As a response to biotic stress, the hosts induce hormonal signals, so high levels of abscisic acid (ABA), indole-3-acetic acid (IAA), jasmonic acid (JA) and salicylic acid (SA) are found [58]. However, this response favors mycelial growth of the fungus, hence hormonal imbalance is established [11,27].

Chlorophyll a and b levels were lower in the inoculated host from day 3 after infection (DAI), with significant differences at 35 DAI and 61 DAI for chlorophyll a, and 7–61 DAI for chlorophyll b, consequently reducing photosynthetic rates [27].

Studies indicate that in the first stage of the disease, the hosts use nonspecific mechanisms to eliminate the fungus [27]. This is inferred from the increase of alkaloids, phenolic compounds, and tannins.

In the face of unspecific and ineffective host defenses, the fungus develops and activates the second phase of WBD, characterized as green broom (Figure 4). At this stage, the host counterattacks by upregulating defense genes that encode protective enzymes [11], and ROS production is still high. There is high production by the fungus of proteases, pectinases, amylases (leading to a reduction in the starch content), glucanases, lipases and cellulases, characterizing the green broom phase of the disease. High levels of nitric oxide (NO) are detected at this stage [15]. There is also an accumulation of hexose in the apoplast [19]. The fungus starts to use starch as a carbon source for its development and lipid peroxidation (MDA) occurs [27]. In this phase of the biotrophic mycelium, the expression occurs at low levels of MpNEP-2, a necrosis and ethylene-inducing protein [33]. However, this expression progresses to the necrosis phase of the disease.

The host’s senescence characterizes the transition to the dry broom phase of the disease where necrosis of infected tissues occurs. In this phase, there is high expression of the necrosis-causing protein, NEP, and consequently high levels of ethylene [33]. In the necrosis stage, different authors have reported the availability of soluble nutrients derived from dead cells of the host, accumulation of SA, phenyllactic acid and mandelic acid [59], low levels of chlorophyll and starch in the hosts, and constant action of MOX and PME of the fungus, favoring the production of basidiomata, completing the disease cycle [19,54], as shown in Figure 5.

### 3.4. Molecular Markers in the Host’s Mechanism of Action

The use of molecular markers as a study strategy was identified in 20% of the manuscripts (Figure 2A). The studies with molecular markers were entirely aimed at identifying polymorphisms associated with resistance to the fungus *M. perniciosa* in the genome of the hosts (Appendix A).

All told, 12.8% of the works identified molecular markers in the genome of the hosts, of which 11.0% explored the genome of *Theobroma cacao* and 1.8% analyzed the genome of *Theobroma grandiflorum*.

Among the classes of molecular markers, various techniques were used to identify simple sequence repeats (SSR), randomly amplified polymorphic DNA (RAPD), SSR associated with expressed sequence tag (EST-SSR), single-nucleotide polymorphism (SNP), and amplified fragment markers and length polymorphism (AFLP). Of the 14 studies that used molecular markers, these classes represented 42.9%, 14.3%, 28.6%, 28.6% and 7.1% of the cases, respectively. We observed some examples where more than one class of molecular markers were used in the same study.

Molecular markers were used in contrasting genotypes in terms of resistance and susceptibility, except for *Theobroma grandiflorum* genotypes. In the host *Theobroma cacao*, 289 molecular markers were identified, regardless of the class, associated with resistance to witches’ broom, with these molecular markers being located in the regions of chromosomes I, IV, VII, VIII, X, and mainly in chromosomes VI and IX (details and references in Appendix A).

The most informative classes of markers for *M. perniciosa* hosts were EST-SSR and SNP, applied for the identification of resistant genotypes.

### 3.5. Protein Profile of M. perniciosa and Its Hosts

The protein profile of the *M. perniciosa* pathosystem and its hosts was traced in 33% of the 109 studies. Of these, 17.4% focused on understanding the stages of development and the mechanism of fungal infection from the proteome of basidiospores, non-germinated basidiospores, biotrophic and saprotrophic mycelia, as well as basidiocarp regions.

A total of 15.6% of the studies traced the protein profile of the hosts in their interaction with *M. perniciosa*. In this context, 94.1% revealed the proteome from *Theobroma cacao* (resistant and susceptible genotypes) and 5.9% identified the proteome from *Nicotiana benthamiana*. The identification of host proteins was aimed at meristematic regions, seeds, stems, phylloplane, apoplasts and suspended cells (details and references in Appendix A). Some studies, in addition to identifying the proteins, also identified the differential regulation and its functions, both of the hosts and fungus (details and references in Appendix A).

In Figure 6A,B, it is possible to observe that *M. perniciosa* activates a protein arsenal to infect its hosts. In spores, proteins related to energy metabolism, pathogenicity, structure and cycle and oxidoreductase are mostly up-regulated, with the exception of some energy metabolism proteins, such as initial translation factor, glyceraldehyde-3-phosphate dehydrogenase, inhibitor of serine protease, nucleoside diphosphate kinase, 60S ribosomal protein and small ribosomal subunit, which are downregulated (Figure 6A).

In the mycelium, the number of proteins upregulated was greater, involving crucial biological functions for the fungus at this stage (Figure 6B). Stress response, energy metabolism and pathogenicity were the most frequent biological functions identified in fungal proteomes in different studies. Proteins related to transport, cell structure and cycle, oxidoreductase, autophagy, macromolecule metabolism, cell wall and binding, and nitrogen metabolism were also identified as causing differential regulation in the fungus x host interaction (Figure 6B and Appendix A).

To resist witches’ broom disease, hosts express proteins associated with different biological functions, mainly those related to defense, oxidative stress, antifungal activity, and energy metabolism (Figure 6C and Appendix A). For each biological function, the identified proteins show different patterns of regulation, mostly upregulated.

In 17.4% of the studies that traced the *M. perniciosa* proteome, a total of 142 proteins were identified (Appendix A). Of these, 77 were submitted to interaction analysis in the STRING database, where only interactions with the established confidence parameters were found in 70 proteins. Of the latter, nine were repeated proteins, leaving 61 proteins for the construction of the protein-protein interaction (PPI) network (Figure 7). Proteins that were not analyzed in the STRING did not meet the criteria established in the methodology (Section 3). Briefly, they were not found by the search engines or had identity <90%.

The physical or functional interactions between the proteins under study are represented graphically and mathematically in the protein-protein interaction network of Figure 7. It is possible to quantify the significance of a protein within a system using certain parameters, such as centrality (betweenness and node degree), modularity (clustering), and others. In this way, betweenness centrality quantifies the number of times a given node acts as a bridge. Proteins with an above average betweenness value have high interaction with other proteins and are called bottleneck proteins. The node degree represents the number of connections that traverse a single node. Therefore, proteins with a node degree value above the average are called hubs and have an important regulatory role in the network [60].

The protein network of Figure 7 contains 1665 nodes (proteins), 20,378 connectors and 13 clusters. Three hundred and nine network proteins are bottlenecks, 592 are hubs, and 250 have both characteristics. Of the 61 proteins from which the network was built, 26 act as both hubs and bottlenecks.

The third cluster contains the highest number of elements, with 621 proteins. These proteins are mainly related to organic acid metabolic processes, generation of precursor metabolites and energy, and oxidation-reduction processes with the most reliable *p*-values (Figure 7; Appendix A). Other processes represented within the network are protein localization (cluster 2) and translation (cluster 1).

Proteins with high average betweenness and node degree values are considered to have an important regulatory role within the network. The three proteins with the highest betweenness values of the entire network are glyceraldehyde-3-phosphate dehydrogenase (phosphorylating) (E2LX05, cluster 3), AAA domain-containing protein (E2L7U8, cluster 2) and pyruvate kinase (E2L4K9, cluster 3). These correspond to proteins reported in works accepted for synthesis of results. In turn, the three proteins with the highest node degree values are KH type-2 domain-containing protein (E2L542), pyruvate kinase (E2L4K9, cluster 3) and S5 DRBM domain-containing protein (E2LEU1). Glyceraldehyde-3-phosphate dehydrogenase (phosphorylating) (E2LX05, cluster 3) has the fourth highest node value.

## 4. Discussion

### 4.1. Brazil Leads in Production of Knowledge about M. perniciosa

The main host of the fungus *M. perniciosa* is *Theobroma cacao*, in terms of economic value. This host is an arboreal, perennial plant that prefers a tropical climate. Upon infecting the host, the fungus causes witches’ broom disease (WBD).

WBD was first reported in Suriname in 1895. At that time Brazil was the second-largest producer of cocoa beans in the world. Although there were records of the endemicity of this disease in the Amazon region since the 19th century, including the northern region of Brazil, it was only in 1989 that the disease was detected in cocoa plantations in southern Bahia state. Since then, Brazil’s production has declined to seventh position in the world [61].

Since 1959, studies have been conducted seeking to describe and understand WBD. Brazilian researchers have focused on the search for varieties resistant to the disease as well as on the understanding of the interaction between *M. perniciosa* and its hosts, producing knowledge at the level of functional genomics, including the proteome. These efforts have put the country in the leading position, with the largest scientific production on *M. perniciosa*, according to the data of this review, accounting for approximately 80% of the studies (Figure 2B,C). According to the data collected, Brazil has contributed local studies and international collaborative research, such as with French researchers (1.8% of the studies) and American researchers (6.4%) (Figure 2B).

The results indicate that the studies between 1959 and 2006 were based on strategies of classical genetic improvement to obtain resistant cultivars. With the results of these studies, Brazilian cocoa farming began showing signs of recovery, although this was not enough, because the new cultivars at that time had a narrow genetic base, i.e., they descended from a single source of resistance, the Scavina-6 variety [4]. In addition, the pathogen has high genetic variability between generations and outgrows host resistance. With technological advances, studies applying sequencing technologies to investigate the fungus and its interaction with hosts revolutionized the understanding of their molecular biology [10,11,17,18,25,26], explaining the significant increase in publications from 2007 onwards (Figure 2).

The sources of the publications on the molecular biology of *M. perniciosa* and its interaction with hosts allow inferring the types of journals that researchers seek to publish their studies. The ten journals with the highest number of publications are identified in Figure 3A. According to Barrios et al. [62], the set of journals is organized in descending order in relation to the production of studies on a given theme, from this, one can identify a group of journals that deal basically with a theme, as done in the present study.

The production of scientific knowledge about the molecular biology of *M. perniciosa* in Brazilian research institutions is impressive. Most of these institutions collaborate with each other, which strengthens this field of study in the country and helps forge partnerships among researchers (Figure 3). Thus, the range of information generated from these different studies is expanded, offering opportunities for a better understanding of the pathosystem, with emphasis on *Theobroma cacao* and *M. perniciosa*.

### 4.2. The Peculiar Battle of a Hemibiotrophic Fungus and Its Hosts

*Moniliophthora perniciosa* is a hemibiotrophic basidiomycete and the causal agent of WBD [63]. Plant diseases caused by fungi with hemibiotrophic life cycle present a brief asymptomatic biotrophic stage, followed by necrosis of the plant, as exemplified by the fungi *Magnaporthe oryzae* and *Colletotrichum graminicola* [64,65]. However, the life cycle of *M. perniciosa* is peculiar, according to the studies covered in this review that have addressed physiological, morphological, and biochemical aspects.

The life cycle of *M. perniciosa* is characterized by a symptomatic biotrophic phase of long duration (lasting more than 60 days in the living tissues of the hosts). Initially, favorable environmental conditions, such as optimal humidity, temperature, and wind strength, facilitate the dispersal of unicellular basidiospores, which adhere to rapidly growing meristematic tissues. The germination time of basidiospores is different between resistant and susceptible genotypes, being shorter (2 h) in the former, and longer in the latter (4 h). In this initial phase of the disease and fungal life cycle, an enzymatic arsenal and mechanical traction are aroused by the fungus to achieve successful penetration and morph into the biotrophic phase [50].

In the biotrophic phase of *M. perniciosa*, still with uninucleate hyphae, the fungus invades the plant tissue and grows in the intercellular region of the host, because unlike other hemibiotrophic phytopathogens, *M. perniciosa* does not yet have structures for nutrient absorption, such as invasive hyphae or haustoria. In turn, the hosts present hypertrophic and hyperplastic anomalous branches and parthenocarpic fruit formation, morphological changes that Mondego et al. described as characteristic of this phase of the disease. In this case, *M. perniciosa* depends on the nutrients available in the apoplast of the plant tissue, spending a period of 30 to 60 days in this region [10].

The apoplast is the first cellular space that the fungus appropriates. Little is known about the molecular tools that the host makes available, and the fungus appropriates to overcome this barrier. None of the studies selected in this review characterized at the molecular level the apoplast of *M. perniciosa* hosts when infected. Regarding the apoplast-fungus fluid interface, only Barau et al. investigated the dynamics of carbon in the apoplast fluid in order to start the end of the biotrophic phase. They observed that glucose and fructose levels were higher in the apoplastic fluid of infected plants in the first 25 days of WBD development, and this increase in glucose and fructose was correlated with a decrease in sucrose levels, suggesting increased activity of invertases [19]. Other authors pointed out that this increase in sugars in the host may represent a plant response to infection [27]. Furthermore, it is believed that *M. perniciosa* manipulates the host’s metabolism to increase the availability of nutrients from the apoplast, ensuring the maintenance of its life cycle [11].

The systematized data from the studies indicated that *M. perniciosa* can grow and develop in the biotrophic phase, employing different plant carbon sources, mainly sucrose, fructose, glucose, and glycerol, the latter produced as a by-product of lipid degradation [11]. Another carbon source found to be important is methanol. The studies reported that methanol can be an optional carbon source during the biotrophic phase of *M. perniciosa*. According to these studies, carbon obtained from hosts regulates the developmental transitions of WBD [9,11,13,19,54]. Other hemibiotrophic fungi, for example, those of the genus *Colletotrichum*, also use glycerol as an energy source for nutrient transfer from infected plants [66]. On the other hand, the host uses nonspecific mechanisms, such as an increase in alkaloids, phenolic compounds and tannins, to try to eliminate the fungus, but without success, since the disease is not avoided, although these compounds can inhibit the germination of basidiospores and cause alteration in the germ tube morphology of the biotrophic phase of *M. perniciosa* [27]. To understand the specificity of the modulation of these secondary metabolites in the host—*M. perniciosa* interaction Scarpari et al. and Gesteira et al. warn about the need for additional biochemical studies, as well as detailed analysis of gene expression of these metabolic routes, to contribute to the understanding of the transition mechanisms of the fungus from the biotrophic to the necrotrophic phase and relate them to the biochemical changes that occur in the broom stage green [27]. In the literature, few studies address the biochemical interaction of *M. perniciosa* and its hosts. Only 30% of the works with this strategy are systematized here, confirming the existence of a gap in the understanding of secondary metabolites and whether this response comprises a specific mechanism or a plant response to any biological stressor.

Meinhardt et al. performed in vitro studies of *M. perniciosa* and observed that after 60–90 days the fungus develops connecting clamps and dicariotized hyphae to invade the intracellular space, and that the rapid change to binucleated mycelia with connecting clamps is characteristic of the necrotrophic phase of this fungus [9]. This change from uninucleate to binucleated mycelia occurs without previous reproduction among compatible individuals, since *M. perniciosa* is a primary homothallic fungus [10].

Calcium oxalate crystals have been shown to play a role as a virulence factor in the fungus, affecting the success of *M. perniciosa* infection in the host. On the one hand, the crystals that appear in the necrotrophic mycelium are formed from calcium ions removed from the pectin of the host cells, facilitating their degradation [51,54]. On the other hand, according to Ceita et al., intracellular growth is accompanied by the development of calcium oxalate crystals, which were found in susceptible uninfected plants, and increased in number after infection with *M. perniciosa* [12]. In resistant plants, these crystals were not observed, so the authors proposed a new role for calcium oxalate of signaling susceptibility to WBD. Furthermore, Dias et al. observed that the resistant genotype accumulated fewer calcium oxalate crystals and these dissolved in the early stages of infection, giving way to the accumulation of H_2_O_2_ [56].

Most studies have focused on understanding the behavior and interaction of the fungus with its hosts in the biotrophic phase. Few have examined the necrotrophic phase, perhaps because of the goal of mitigating the symptoms of the disease while still in the biotrophic phase, and because of the data indicating that the green broom phase is a point of compromise in disease progression [11]. However, researchers who have investigated the necrotrophic phase admit that the arrival at the dry broom stage is a slow and adaptive process to the hostile environment, where the host expresses its arsenal of defense. According to Pungartnik et al., a gradual exhaustion of the biochemical mechanisms of the fungus occurs, while necrotic action is induced, and rapid intracellular growth takes place [14]. At that stage, *M. perniciosa* already has dikaryotic hyphae penetrating the host, generating necrosis of plant cell tissues, i.e., parallel to the death of the infected plant tissue, which occurs after 90 days of infection. *M. perniciosa* completes its cycle in an increasingly inhospitable environment, with alternating wet and dry periods. The fungus produces basidiomes that release basidiospores, restarting the WBD cycle [14,67,68,69].

Sena et al. studied the development of *M. perniciosa* in resistant and susceptible genotypes of *Theobroma cacao*. They observed the same pattern of fungal invasion in both genotypes, but at a slower rate in the resistant genotype than in the susceptible one. In the resistant genotype, the beginning of green broom was evident only as of week 5, while in the susceptible genotype this condition was observed from week 3 after inoculation. In addition, the resistant genotype produced smaller brooms, less hypertrophied cortex and phloem, shorter and thinner intercellular hyphal segments, and overall lesser pathogen proliferation [50].

Barau et al. reported a curious feature of WBD in *Theobroma cacao*: when its tissues are necrotized and dead, they remain attached to the plant for a long time [19]. It is believed that this behavior is advantageous to the fungus because it has first access to the biological resources contained in the dead tissues and neutralizes any competition with other microorganisms. Moreover, remaining in dead tissues increases the probability of spore dispersion, including to nearby living tissues, guaranteeing its life cycle in an efficient and peculiar way. This intriguing characteristic of *M. perniciosa* in dead cocoa tissues was also reported by Purdy and Schmidt [67], Scarpari et al. [27] and Meinhardt et al. [9].

Among the research strategies used in the studies reviewed here, analysis of molecular markers was the most-used technique, more so in the hosts than the fungus *M. perniciosa*. Among the different classes of molecular markers, there were five leading ones (as cited in the results section), only pertaining to two hosts of the fungus, *Theobroma cacao* L. and *Theobroma grandiflorum*. In general, these studies aimed to characterize molecular markers to identify genetic variability and resistance genes in the hosts. These studies were published between 2005 and 2011, when the genome of the pathosystem (fungus x host) had not yet been published. Indeed, the molecular markers for *Theobroma grandiflorum* were only described in 2016 [70]. This shows the wide interest in this technique, so SSR or SSR-EST markers are well characterized for some cacao genotypes [71,72,73,74,75,76,77,78] (Appendix A).

When the overall goal of identifying molecular markers in hosts is to develop resistant or high-yielding varieties, authors claim that this can be accelerated using marker-assisted selection (MAS), which, associated with the analysis of quantitative trait loci (QTL) that control resistance genes, can be used to select varieties of interest [79,80,81]. However, these associated methods lack refinement, since they rely on intensive genotyping with SNP molecular markers, either via next-generation sequencing (NGS) or by microarrays. However, they are more robust [80,82,83].

Santana et al. analyzed isolates of *M. perniciosa* using RFLP [4]. However, studies such as these this are scarce, even when prioritizing other classes of markers, unlike host studies. For *M. perniciosa*, SSR molecular markers were first characterized by Gramacho et al. [84] and Silva et al. [85] in different isolates. The RAPD molecular markers were also investigated by Andebrhan and Furtek [86]. However, considering the co-evolution of *M. perniciosa* and hosts, it is necessary to investigate the construction of a map of molecular markers for the pathogen, perhaps using the class of SNPs, since they identify single-base polymorphisms.

### 4.3. Structural Genomics of the Causal Agent of Witches’ Broom

Descriptions of the genome sequences of the causal agent of WBD, *M. perniciosa*, published in recent years [10,25,26], along with the genome of one of its main hosts of socioeconomic importance, *Theobroma cacao* [17,18,87], and the dual transcriptome (*M. perniciosa* and *Theobroma cacao*—Atlas Transcriptome) revealed by Teixeira et al. [11], opened the way for application of structural genomics to obtain more detailed molecular information about this pathosystem. Another important development is genetic manipulation through techniques based on CRISPR/cas9 and epigenetic systems, to shed light on problematic organisms such as *M. perniciosa*.

To complete its life cycle, *M. perniciosa* wages a molecular battle with its host, activating or repressing genes crucial for its success. Genes associated with pectinolytic metabolism are positively regulated in the initial phase of the disease [11], confirming the colonization of *M. perniciosa* in the host’s mid-layer, rich in pectin, as well as the degradation of this polysaccharide in the initial green stage. In entirely necrotrophic fungi, the pectin degradation activity is more common and well characterized, as is the case of the fungus *Botrytis cinerea* [88]. This reiterates how peculiar the biology of *M. perniciosa* is.

In the process of degradation and demethylation of pectin, methanol is released, used as an energy source, and enters a cascade to form carbon dioxide. In this process, one of the byproducts is glyceraldehyde 3-phosphate, which can be a precursor for the synthesis of oxaloacetate. Oxaloacetate removes calcium ions from the pectin structure, forming calcium oxalate and allowing further enzymatic degradation of pectin by plant cell wall degradation enzymes [54,89]. This event is particularly evident in susceptible hosts. Previously, studies predicted the formation of CaOX crystals only in the early stage of the disease [12], but its production has also been observed in necrotrophic mycelia of *M. perniciosa* [51,56].

The fungus initially grows in the intercellular space of the plant cell, invading the apoplast. In this region, during the proliferative phase of the disease, Barau et al. found a significant increase in glucose and fructose levels in parallel with a reduction in sucrose levels, suggesting upregulation of genes related to cell wall invertases, as well as the accumulation of fungal and plant enzymes for the maintenance of apoplastic hexose from sucrose breakdown [19]. According to these findings, infection initially leads to the accumulation of hexose, followed by consumption of the soluble carbohydrates in the apoplastic fluid.

Teixeira et al. reported that *M. perniciosa* seems to manipulate the metabolism of the host to increase the availability of nutrients in the apoplasts, triggering the formation of monokaryotic hyphae, which favors the availability of more nutrients to the fungus [11]. This characteristic supports the upregulation of fungal genes, identified to encode oligopeptide and monosaccharide transporters, proteases and asparaginase in the initial phase of the disease (green broom). This process degrades apoplastic proteins and enables use of the final peptides [11]. Host manipulation for nutrient acquisition has also been observed in other phytopathogens, such as *Colletotrichum gloeosporioides* f. sp. malva, which uses glycerol for nutritional transfer [66].

Still at the green broom stage, genes encoding antioxidant enzymes such as superoxide dismutase and catalase are upregulated in *M. perniciosa* according to the studies reviewed here. These enzymes detoxify ROS, such as superoxide anions (O_2−_), hydroxyl radicals (OH) and hydrogen peroxide (H_2_O_2_) and are produced by plants to prevent pathogen invasion and increase stress tolerance [90]. In contrast, hosts also activate the gene arsenal of their antioxidant system to develop metabolites and enzymes, such as peroxidases, that prevent entry into the fungus’ intracellular space [91].

In the Transcriptome Atlas of Teixeira et al. [11], the pathogenesis related (PR) genes of *M. perniciosa* are among the most highly expressed genes during interaction with the host *Theobroma cacao*, with emphasis on MpPR-1, which acts to detoxify lipid toxins produced by the host, thus protecting the fungus. This protein belongs to the class of cysteine-rich secretory proteins, including antigen 5 and pathogenesis-related 1 (CAP), which according to Darwiche et al. are implicated in fungal virulence and immunosuppression [92]. Another class of microbial PR whose genes are over-represented in the green broom stage are the protein cerato-platanins (CPs). CPs prevents the plant’s fungal recognition receptors from detecting it, impairing host defense, and favoring success of the biotrophic stage [11].

Also, when studying biotrophic mycelium, Thomazella et al. identified increased expression of the *Mp-aox* gene accompanied by high mitochondrial alternative oxidase (AOX) enzyme activity in *M. perniciosa* during the infection process [15]. This suggests the important role of this enzyme during the biotrophic phase. AOX protects the fungus against the harmful effects of mitochondrial respiratory chain byproducts, among them nitric oxide (NO), and further regulates the transition to the neurotrophic phase of WBD. Researchers have hypothesized that although hosts, in response to fungal attack, produce high concentrations of H_2_O_2_, triggering programmed cell death (PCD) of infected tissue cells as a defense mechanism, this actually favors the fungus since it makes more nutrients available by stimulating the phase change (biotrophic–necrotrophic) of WBD [12].

A few studies have identified gene expression of *M. perniciosa* in the necrotrophic phase. Examples are Lanver et al. and Basse et al., who studied the phytopathogen *Ustilago maydis*, a biotrophic fungus with a dimorphic lifestyle [93,94]. In its biotrophic phase, it is not pathogenic, unlike *M. perniciosa*. This indicates the need to understand the action of the fungus as early as possible in its biotrophic phase, to prevent development of the necrotrophic phase.

Key genes related to the necrotic phase of the disease are well described as to their expression in the infection process. Examples are the genes that code for the necrosis and ethylene inducing proteins (NEPs). These proteins act in the extracellular medium and were first identified in the fungus *Fusarium oxysporum*, which also has the ability to induce necrosis in its hosts [33,95,96,97]. The onset of necrosis is parallel to the death of infected host tissues [69].

Overall, infection by *M. perniciosa* causes a general derangement in host metabolism and culminates in the expression of key genes in this molecular battle. The information generated by different molecular studies sheds light on the regulatory networks that control fungal pathogenicity, as well as host defense, by identifying the changes in gene expression regulation that occur in this battle. The results directly impact agricultural production and food security.

Large-scale gene expression analyses of plant-pathogen interactions are of great relevance to unveil the molecular basis of a specific disease. However, when it comes to *M. perniciosa* and its hosts, studies focused on genomic editing using CRISPR/cas9 systems as well as the action of epigenetic mechanisms on gene regulation have not yet been published.

Some studies have already described CRISPR-Cas9 genome editing technology for fungi of major socioeconomic importance, such as *Magnaporthe oryzae* [98], *Neurospora crassa* [99], *Trichoderma reesei* [100] and *Ustilago maydis* [101]. In the case of *Ustilago maydis*, for example, the authors aimed to efficiently disrupt functionally redundant target genes in the corn phytopathogen [101].

This same pathway has been paved for some of the main hosts of these phytopathogens. An example is *Oryza sativa*, which is affected by Brusone’s disease in rice, caused by the fungus *Magnaporthe oryzae*. The study of [102] revealed that when the *nlr* gene, which confers resistance to the disease by rice, was knocked out via CRISPR-Cas9, resistance in transgenic plants was partially reduced. Unlike rice, for *Theobroma cacao*, the main host of *M. perniciosa*, there are no studies of the CRISPR-Cas9 system conferring resistance, since there is no established method.

The regulation of gene expression under the effects of epigenetic mechanisms is another field not yet explored involving the *M. perniciosa* pathosystem and its hosts. The understanding and manipulation of resistance responses of plants with great agronomic importance affected by biotic and abiotic factors from an epigenetic standpoint is already well described for some species [41,103,104,105]. In this sense, there are no reports of *Theobroma cacao*, the main host of *M. perniciosa*, so there are no methods that aim to identify epigenetic mechanisms associated with resistance to the fungus. There is hence a need to clarify the action of epigenetic mechanisms in the regulation of genes that confer pathogenicity in order to understand their action in the different stages of WBD, and, if possible, to manipulate them. In a study of *Ustilago maydis*, the authors hypothesized that epigenetic control through histone acetylation would act to control the transcription of genes related to pathogenesis, virulence and growth of the fungus [106].

Differences in some molecular responses between resistant and susceptible genotypes of *Theobroma cacao* have been reported. Almost all of them are related to the antioxidative system of the plant, such as higher amounts of H_2_O_2_, oxalic acid and/or ascorbic acid, induction of the genes for oxalate oxidase (G-OXO), germinal-type oxalate oxidase (Glp), and dehydroascorbate reductase (Dhar) [12,56], as well as higher activity of the APX enzyme [57,91] in resistant genotypes in the interval from 0 h to three days after infection with *M. perniciosa*. Another interesting situation, discussed previously, is that resistant genotypes produce much smaller amounts of calcium oxalate crystals in the basal state and these are dissolved during the initial phase of infection, leaving in their place H_2_O_2_ [56]. This information indicates that the key feature of the resistant genotypes is their detoxification mechanism from the first hours of infection.

### 4.4. The Hidden Biotechnological Potential of M. perniciosa Pathosystem Proteins

The publication of the genome of *M. perniciosa* and its main host, *T. cacao*, has allowed the integration of proteomics studies, helping resolve the puzzle about WBD. Unlike the genome, the proteome reflects biological processes triggered under different physiological conditions or pathological states [107].

Proteins expressed during phytopathogenic interactions have been successfully identified through proteomic analyses [108]. Furthermore, biochemical and structural characterization of proteins, whether inferred from the genome or identified in the proteome, contributes to unravel the molecular mechanisms of the battle between *M. perniciosa* and its hosts.

In this regard, 19 *M. perniciosa* proteins of interest have been characterized in vitro (Appendix A): two NEPs [33], four CPs [109,110], 11 pathogenesis-related PR-1 proteins [92], one Acyl-CoA binding protein [111] and one inactive chitinase [55].

One of the biggest mysteries surrounding WBD is the transition from the biotrophic to the necrotrophic phase, because it is still unclear which factor induces this change. Initially, the hypotheses pointed to action peculiar to *M. perniciosa*, so the first proteins to be characterized in vitro were those found in the genome of *M. perniciosa*, similar to proteins from other fungi that had already been shown to cause necrosis in hosts [33].

The proteins MpNEP1, MpNEP2 and MpCP1 induce necrosis in tobacco and cocoa leaves. Although the genes for these three proteins are expressed in the biotrophic mycelium, infected plants show no visible symptoms for many weeks. This suggests that the proteins need a minimum concentration to cause symptoms in tissues [33]. Furthermore, MpNEPs induce ethylene production in the plant long before necrosis symptoms are visible. Although the effects of MpCP1 and MpNEP2 are different, when subjecting leaves to treatment with both proteins, a necrosis effect similar to that found in naturally infected plants occurred [110]. Furthermore, MpNEP2 induced detachment of cell membranes and affected ATP synthesis in tobacco and has also been found to be related to an increase in NO synthesis [112].

Thanks to the publication of the *M. perniciosa* genome, it has been possible to find orthologous gene families that had already been studied in other fungi. One of these families was the cerato-platanins, of which 12 genes were identified. These showed different expression profiles during infection [109]. MpCP1, MpCP2, MpCP3 and MpCP5 proteins bind to chitin fragments according to in vitro characterization. MpCP5 blocks the perception of chitin monomers by the plant, while MpCP1, MpCP2, and MpCP3 facilitate the process of hyphal growth, fruiting body formation, and substrate adhesion. In addition, aggregates of MpCP2 were able to promote cellulose fragmentation as well as contribute to pollen tube formation [109].

Interestingly, in the genome of *M. perniciosa*, orthologous genes from other fungi, and also from plants have been identified, such as PR-1 (pathogenesis-related 1) family proteins, involved in fungal virulence and immunosuppression. From this family, 11 members have been identified, revealing that sterol and fatty acid binding is important in WBD progression [92].

Finally, a MpChi (inactive chitinase) has been shown to function as a putative pathogenicity factor, preventing chitin-triggered immunity by sequestering immunogenic chitin fragments [55].

Hundreds of candidate effector proteins have been identified in silico in the *M. perniciosa* genome [22]. However, it is still necessary to characterize them in vitro and in vivo for a better understanding of the fungal strategy and the WBD process, as well as to identify targets for control of the disease.

Efforts to understand the molecular mechanisms during the development of *M. perniciosa* through proteomics have been carried out in the last decade. Protein profiling of basidiospores has revealed that most of the identified proteins are related to energetic and oxido-reduction processes, in addition to identifying proteins important to hyphal development and branching [113].

In turn, the protein profile of *M. perniciosa* spores in the germination phase revealed proteins associated with fungal filaments, such as septin and kinesin, in addition to positive regulation of oxidative stress-related proteins, such as SOD and catalase, which possibly help in the detoxification of free radicals within the primary hyphae. Virulence factors, such as polyketide synthase and FapR, were also present in the basidiospores and primary hyphae [21].

Two proteomic studies of the necrotrophic phase of the mycelium were identified. One compared the S and C biotypes [20] and the other compared six developmental stages of the fungus according to mycelium color and development (white, yellow, pink, dark pink; primordium and basidiocarp) [23]. In the first study, proteins involved in virulence, pathogenicity and stress response were identified abundantly in all cultures [20]. In the second one, proteins related to IAA metabolism, cell proliferation and cytoskeleton activity were identified most abundantly in the white mycelium phase, suggesting they are required for mycelial development. In the dark pink phase, proteins were identified related to vesicular transport processes mediated by small GTPase signaling, cell wall biogenesis, stress response and response to nutrient deficiency, which were also associated with the primordium phase [23].

Proteomic studies of *M. perniciosa* subjected to treatments with leaf washes and with a PR-10 of *Theobroma cacao* were found. Basidiospores of *M. perniciosa* germinated in the presence of leaf washes from contrasting genotypes of *Theobroma cacao* showed a distinct protein profile [114]. The authors highlighted the reduced ATP synthase of the germinated basidiospores in the leaf wash of the susceptible genotype, suggesting a shift from aerobic to fermentative metabolism.

On the other hand, *M. perniciosa* hyphae treated with TcPR-10 had positive regulation of proteins related to stress response, detoxification, autophagy, and maintenance of fungal homeostasis [34].

A protein–protein interaction network was constructed from the *M. perniciosa* proteins reported in some of the papers (Figure 7 and Appendix A). The centrality values indicated that among the most important proteins for the regulation of the network are glyceraldehyde-3-phosphate dehydrogenase (phosphorylating) and pyruvate kinase, both of which are in the cluster related to organic acid metabolism and are specifically involved in glycolysis, indicating that obtaining energy by this pathway is a key process for the development of the fungus and the disease. It has already been shown that an increase in the amount of glucose in the infected plant occurs in the initial weeks [19,27], so it is an important energy source for the fungus during the progression of WBD.

Cluster 2 had the second most proteins, related to transport, localization, and the cell cycle. This cluster included proteins involved in the formation of structures during cell division, consistent with the development of the fungus, considering the change from monokaryotic to dikaryotic form in the transition from green broom to dry broom [14]. Interestingly, the protein with the most connections within this cluster, an AAA domain-containing protein (E2L7U8), has an unknown function.

Another interesting biological process in the PPI network was the metabolism of phospholipids, formed by proteins that are directly related to phosphatidylserine decarboxylase. This enzyme is involved in the synthesis of phosphatidylethanolamine, a mitochondrial phospholipid whose depletion in yeast causes respiratory dysfunction, defects in the assembly of mitochondrial protein complexes and loss of mitochondrial DNA [115,116]. Therefore, it can be studied as a molecular target to control the development of *M. perniciosa*.

With regard to the hosts, 10 *Theobroma cacao* proteins of biotechnological interest have been characterized in vitro: four cystatins [117], one TcPR-10 [118], one β-1,3-1,4-Glucanase (TcGlu2) [119], one PR-4b [120], and one osmotin [121]. All of them have been shown to have a fungicidal effect on *M. perniciosa*. TcCYSPR04 [122] was also characterized in vitro but showed no significant fungicidal effect. Another protein from *Theobroma cacao*, the Kunitz-type trypsin inhibitor, was tested in vitro for its biotechnological potential and efficiency, but only as a potential larvicide against *Helicoverpa armigera* [123].

The phylloplane is considered the first battleground for cocoa against *M. perniciosa*, and the resistant genotype CCN51 has an index of short glandular trichomes two times higher than the susceptible genotype, Catongo [124]. Water-soluble compounds from the phylloplane of *Theobroma cacao* have demonstrated an inhibitory effect on germination of spores of the fungus [114]. The proteomics of the phylloplane of the cocoa genotype resistant to WBD revealed proteins related to defense, synthesis of defense metabolites and the metabolism of nucleic acids, demonstrating that proteins and water-soluble compounds secreted to the phylloplane of cocoa participate in the defense against pathogens. In this regard, the expression of a phylloplanin, identified in the genome of *Theobroma cacao* (TcPHYLL), was demonstrated in leaf trichomes of transformed tobacco plants [125].

Recently, the protein profile of contrasting genotypes for resistance to WBD inoculated with *M. perniciosa* and its controls was studied. This revealed that pathogenesis-related proteins (PRs), proteins related to the regulation of oxidative stress and trypsin inhibitors, and a strong detoxification mechanism are involved in WBD resistance [126].

Although proteomic studies have helped to clarify various processes of the interaction between *M. perniciosa* and its hosts, no gel-free proteomic studies were found among the articles that met the inclusion criteria of this review. This high-throughput technique could provide more complete information about the development of WBD.

## 5. Conclusions

This review summarized information accumulated in recent years on the molecular biology of *M. perniciosa*, shedding light on the data produced through systematization, and allowing both a better understanding of the fungal pathosystem and the identification of gaps in the literature.

Although in recent years invaluable methods and resources have been explored to understand the molecular biology of *M. perniciosa* and fungi-host interactions, it is still important to determine how the biotrophic phase is maintained in *M. perniciosa*, and at the molecular level, to ascertain how their hosts contribute to the end of this phase of WBD. Comprehending the transition from biotrophic to necrotrophic phase is crucial for control of the disease, as well as for the development of resistant hosts.

In addition, the results reviewed can allow for the application of cutting-edge methods, such as the CRISPR editing system to knockout genes of interest or using mechanisms from epigenetic memory, to improve understanding of the main genes involved in the end of this fungal stage and how to manipulate them.

Most of the proteins identified in the different studies have strong biotechnological potential, mainly the fungal effectors. Although these are still poorly characterized, they are crucial to the molecular battle of *M. perniciosa* versus hosts.

Genes upregulated in the first hours of infection with *M. perniciosa* in resistant hosts, whose functions are related to defense processes and resistance to diseases, such as chitinases, CC-NBS-LRR resistance proteins, leucine-rich protein membrane receptors, osmotins, peroxidases, thaumatins and WRKY transcription factors, could be key in WBD resistance.

Genes and proteins involved in carbohydrate metabolism, enzymes that degrade cell wall components, oxidative stress response enzymes, cerato-platanin family genes and proteins, and candidate genes for secreted effectors have been highlighted in several studies as important virulence factors of *M. perniciosa* in the WBD.

In turn, proteins from resistant *T. cacao* genotypes that were positively accumulated during infection were those involved in the response to oxidative stress, such as peroxidases, SOD and APX; proteins involved in defense such as PRs, HSPs and protease inhibitors, as well as proteins that have been shown to act directly in inhibiting fungal growth such as TcPHYLL, cystatins, legumains, TcPR-4b and TcPR-10 are strong candidates for functional biotechnological studies 

Literature mining brings to light the importance of reaching a better understanding of *M. perniciosa*–host interactions and WBD. Therefore, this study identified some of the pieces of this biological puzzle that are missing, and the prospects for finding them. Indeed, by presenting the knowledge about genes, proteins, molecular markers, physiological effects and biochemicals involved in the fungal pathosystem, we were able to identify the gaps the must be filled to allow for effective mitigation of the disease.

## Figures and Tables

**Figure 1 ijms-24-05684-f001:**
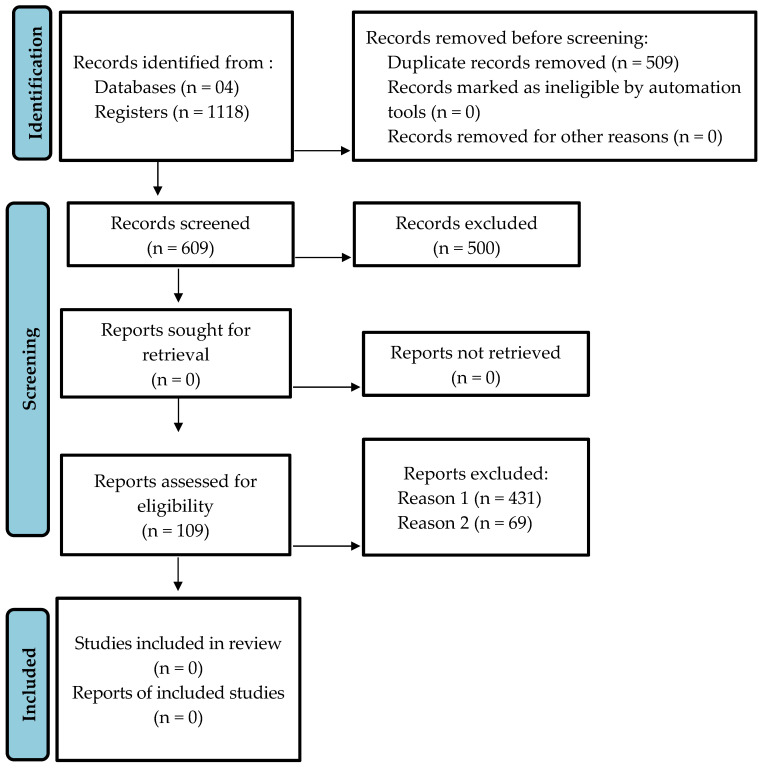
Flowchart with identification and selection of studies related to the molecular mechanisms of interaction between *M. perniciosa* and its hosts, according to PRISMA guidelines. n: number of studies.

**Figure 2 ijms-24-05684-f002:**
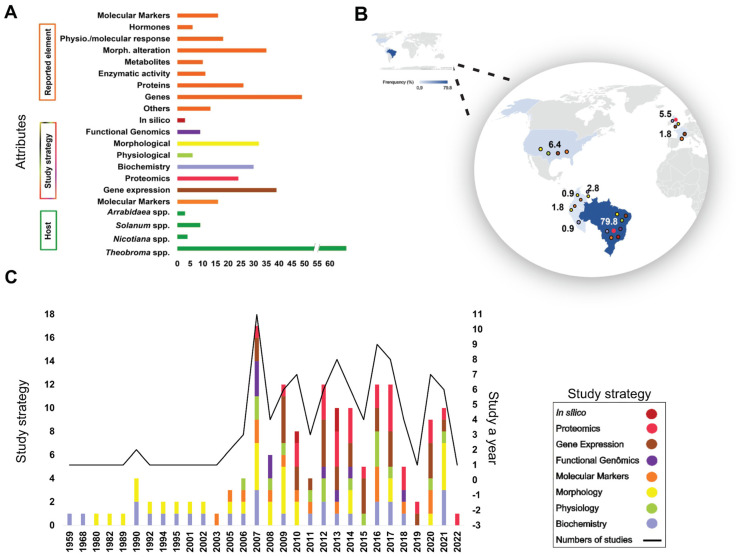
Bibliometric indicators of selected studies of the molecular biology of the interaction between *M. Perniciosa* and its hosts: (**A**) attributes identified in the papers; (**B**) frequency of country studies with their respective study strategies; and (**C**) frequency of studies per year. Physio.: Physiological; Morph.: Morphological.

**Figure 3 ijms-24-05684-f003:**
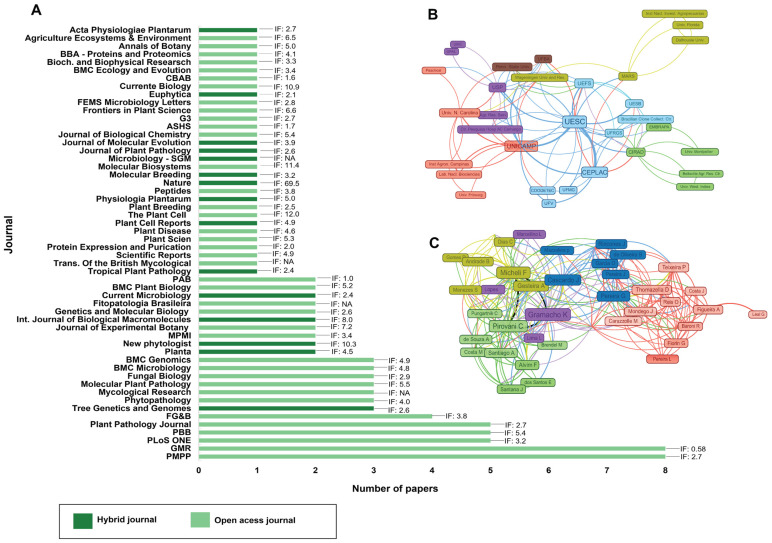
Production and dissemination of scientific knowledge on the molecular biology of *M. perniciosa* and its hosts: (**A**) distribution of studies by journal; (**B**) collaboration network among the authors’ research institutions; and (**C**) collaboration network among authors. Nodes with the same colors in figures (**B**) and (**C**) represent, respectively, institutions and authors grouped by similarity by the Vosviewer software (version 1.16.17). The font sizes of the captions are proportional to the number of publications with participation of institutions and authors. The journals’ impact factors were taken from their own websites in August 2022. IF = Impact factor.

**Figure 4 ijms-24-05684-f004:**
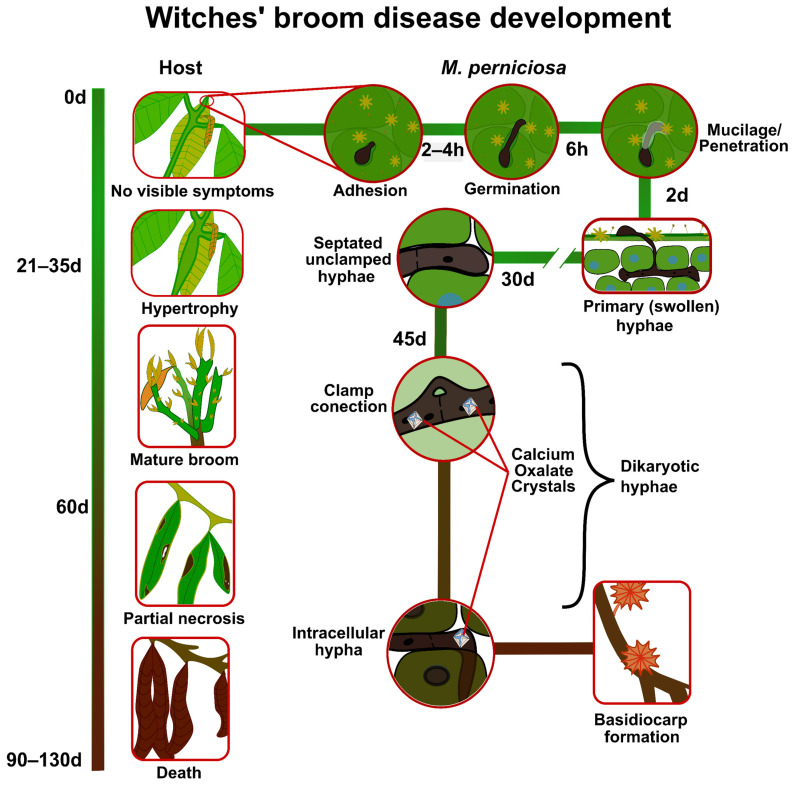
Developmental cycle of witches’ broom disease (WBD) over time in the most studied host (*Theobroma cacao*) of *M. perniciosa* involving the morphological and biochemical aspects of the compatible interaction; d = days; h = hours.

**Figure 5 ijms-24-05684-f005:**
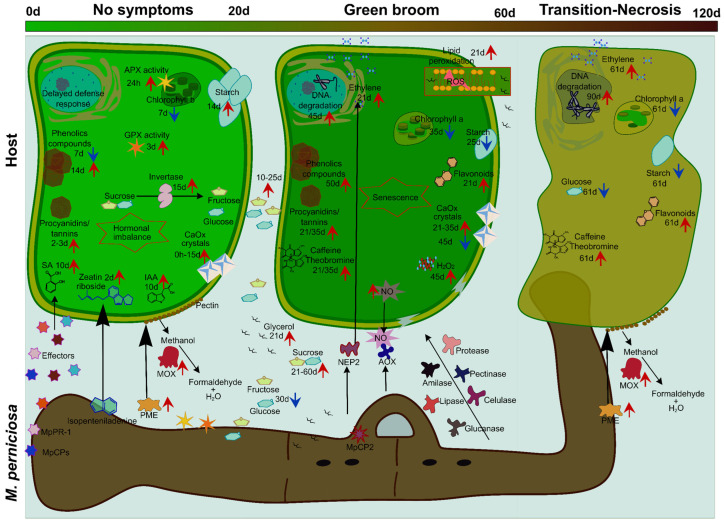
Molecular processes triggered in the *M. perniciosa* x host pathosystem in the development of witches’ broom disease in compatible interaction. The three stages of witches’ broom disease are presented: no symptoms, green broom, and necrosis. Red up arrows represent increase and blue down arrows a decrease of reported substances at the stage or time indicated in days (d) or hours (h). APX: Ascorbate peroxidase; GPX: Glutathione peroxidase; MpPR: Pathogenesis-related protein from *M. perniciosa*; MpCPs: Cerato-Platanins from *M. perniciosa*; PME: Pectin methyl esterase; NEP2: Necrosis and ethylene-inducing protein 2; MOX: Methanol oxidase; CaOx: Calcium oxalate crystals; ROS: Reactive oxygen species; H_2_O_2_: Hydrogen peroxide; NO: Nitric oxide; AOX: oxaloacetate; SA: Salicylic acid; IAA: Indoleacetic acid; PME: Pectin methyl esterase MpCP2: Cerato-platanin protein 2.

**Figure 6 ijms-24-05684-f006:**
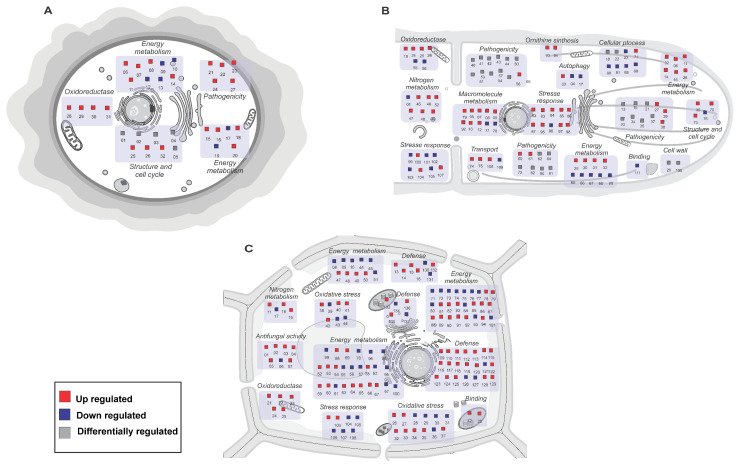
Identified proteins with differential regulation in studies that have used proteomics as an analytic strategy: (**A**) proteins differentially regulated identified in *M. perniciosa* spores; (**B**) proteins differentially regulated identified in the mycelium of *M. perniciosa*; and (**C**) proteins differentially regulated in the host *Theobroma cacao* when infected by *M. perniciosa*. Proteins are grouped according to their biological functions. The numbers represent identification of proteins shown in Appendix A. Differential regulated: in the different experimental conditions of the analyzed studies.

**Figure 7 ijms-24-05684-f007:**
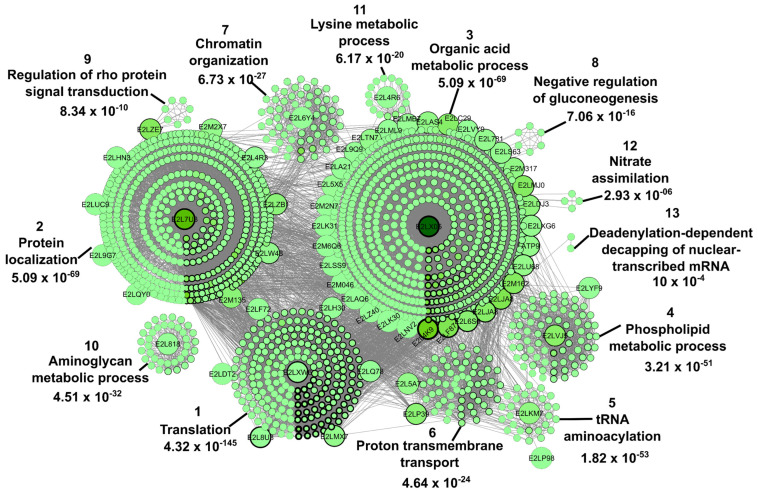
Protein-protein interaction network. Each node represents a protein. The largest nodes represent the proteins from Appendix A. Each numbered group represents a cluster. with an assigned biological process. The betweenness value is represented by the filling color of the nodes, where the lightest color represents the lowest value and the darkest color the highest value. The node degree parameter is represented by the nodes’ border width, where nodes with a thinner border have a lower node degree and nodes with a wider border have a higher node degree value.

**Table 1 ijms-24-05684-t001:** Questions designed for the systematic review.

Questions
1. What are the main research groups studying witches’ broom disease caused by *M. perniciosa*?
2. In which countries is research conducted involving witches’ broom disease caused by *M. perniciosa*?
3. What are the areas of knowledge of publications on witches’ broom disease caused by *M. perniciosa*?
4. What are the hosts of the fungus *M. perniciosa* and the frequency of publications by host?
5. What are the molecular mechanisms induced in the fungus *M. perniciosa* in the infection process or in its development?
6. What are the molecular mechanisms induced in the hosts in the infection process caused by *M. perniciosa*?
7. What are the genes related to resistance (or susceptibility) in the interaction between *M. perniciosa* and its hosts?
8. What are the epigenetic mechanisms involved in host resistance or susceptibility?
9. Which genes are involved in the pathogenicity (virulence) of the fungus?
10. What are the molecular markers associated with host resistance?
11. What sources of resistance have been identified or developed against the fungus *M. perniciosa*?
12. What morphological changes does the fungus *M. perniciosa* undergo in order to succeed in the infection process?
13. Which morphological changes do the hosts undergo in the process of infection caused by the fungus *M. perniciosa*?
14. Which genes and proteins are involved in the *M. perniciosa* x host molecular battle?

**Table 2 ijms-24-05684-t002:** Description of the PICOS strategy used in the systematic review.

Acronym	Definition	Components of the Question
P	Population	*M. perniciosa* causing witches’ broom disease and its hosts
I	Intervention or Interest	To describe the state of the art of the molecular biology of the witches’ broom disease caused by *M. perniciosa*
C	Comparison	The molecular biology of fungus-induced infection and the molecular biology induced by hosts when infected.
O	Outcome	Description of the molecular biology of the interaction between *M. perniciosa* and its hosts, as well as the molecular mechanisms that confer resistance or tolerance to *M. perniciosa*.
S	Type of Study	Review of experimental scientific studies.

## Data Availability

The data presented in this study are available in Appendix A.

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
