# Peer review of "State of the Art of the Molecular Biology of the Interaction between Cocoa and Witches’ Broom Disease: A Systematic Review"

_ijms, 2023, doi:10.3390/ijms24065684_

Round 1
Reviewer 1 Report
Dear Authors,
please find the recommended changes and comments highlighted in the pdf file. The quality of the figures, beside figure 1, is scarce, therefore they need to be uploaded again in a higher quality.
Best Regards.

Author Response
We sincerely appreciate yurs contribution and suggestions regarding the manuscript "State of the art of the molecular biology of the interaction between cocoa and witches' broom disease." (Manuscript ID ijms-2003855)
All suggestions were entirely accepted and highlighted using the “Track Changes” function in the text.
Reviewer #1
- Line 36 – Shaping instead of influencing? - Have been corrected and revised accordingly.
- Line 52 - ?? - Have been corrected and revised accordingly.
- Line 56 – Add space - Have been corrected and revised accordingly.
- Line 77 – understanding OF the molecular - Have been corrected and revised accordingly .
- Line 245 – 248 – Exception made for Figure 1., all Other figures nede na improvement in quality - All figures were sent according to the quality required by the journal and in accordance with the submission guide for authors. The figures have a resolution of 500 DPI, in addition we send the figures in PDF format to avoid loss of quality in the sending system. However, in the version of the manuscript sent by the reviewer, the figures are not at the same level of quality sent to the journal, we are sorry. That's why we certify the quality of the figures.
- Figure 3 number of papers - Have been corrected and revised accordingly.
- Line 363 – Differentially expressed compared to Other phases is meant? - Information has been added to the text better explain results.
- Line 477 – I find accumulated not elegant for meaning up regulated proteins. In addition, up-accumulated sounds like repeated double the meaning. I recommend to switch to up-regulated in the manuscript - Have been corrected and revised accordingly.
- Line 482 – Explain this better - Information has been added to the text better explain results
- Line 764- To be corrected - Information has been added to the text better explain discussion.
Reviewer 2 Report
The manuscript is relevant in the field of molecular level of witches’ broom disease and cacao and presents the molecular level of the pathogen-host interaction on Moniliophthora perniciosa.. However, there are several points need to be considered:
- Theobroma or Theobroma cacao must be presented in the abstract as the main material discussed in the manuscript.
- Avoid personal pronouns in abstract.
- The author should write a more scientific background regarding the existence of chocolate production (lines 27-29).
- Even without visible symptoms of infection, studies have shown that penetration is mechanical and enzymatic (line 306). What kind of enzyme appears in this process?
- There are no citations to statements in lines 302-548.
- What is the role of calcium oxalate at day 45 of infection?
- In 27 studies, it was possible to count 264 genes identified and discussed in different 339 hosts infected by the fungus. These genes were identified in the hosts Theobroma cacao, Theobroma grandiflorum, Solanum lycopersicum L and Nicotiana tabacum, where in most cases transcriptional activity was detected (lines 339-346).
Exactly how many studies and genes authors applied specific for Theobroma?
- Regarding to genes reported in hosts infected with M. perniciosa (lines 347-361) should be presented better, so that it is easier for readers to understand.
- Explain the effect of these compounds (alkaloids, phenolic compounds, tannins, and sugars in the apoplast) during the green broom phase (lines 361-362).
- In conclusion, kindly state clearly the transition phase which is crucial for the control of the disease.
- Kindly improve the Figure captions (Figures 3, 5, 6 and 7).
- Kindly check the broken link (Lines 108-109).
- There are still many old references (more than 5 years and above).
Author Response
We sincerely appreciate yurs contributions and suggestions regarding the manuscript "State of the art of the molecular biology of the interaction between cocoa and witches' broom disease." (Manuscript ID ijms-2003855)
All suggestions were entirely accepted and highlighted using the “Track Changes” function in the text.
Please find our comments on suggestions below.
Sinceriously yours,
Carlos Priminho Pirovani
Reviewer #2
- Theobromaor Theobroma cacao must be presented in the abstract as the main material discussed in the manuscript. - Information has been added to the abstract.
- Avoid personal pronouns in abstract - Have been corrected and revised accordingly.
- The author should write a more scientific background regarding the existence of chocolate production (lines 27-29) - Information has been added to the text better explain of chocolate production.
- Even without visible symptoms of infection, studies have shown that penetration is mechanical and enzymatic (line 306). What kind of enzyme appears in this process? -Information was added.
- There are no citations to statements in lines 302-548 - Citations have been added.
- What is the role of calcium oxalate at day 45 of infection? - Information has been added in the discussion.
- In 27 studies, it was possible to count 264 genes identified and discussed in different 339 hosts infected by the fungus. These genes were identified in the hosts Theobroma cacao, Theobroma grandiflorum, Solanum lycopersicum L and Nicotiana tabacum, where in most cases transcriptional activity was detected (lines 339-346).
Exactly how many studies and genes authors applied specific for Theobroma?- Information has been added.
- Regarding to genes reported in hosts infected with perniciosa(lines 347-361) should be presented better, so that it is easier for readers to understand -The pointed paragraph has been modified.
- Explain the effect of these compounds (alkaloids, phenolic compounds, tannins, and sugars in the apoplast) during the green broom phase (lines 361-362) -An explanation of the role of these compounds during infection was added to the discussion at lines 664-668 and 678 - 682.
- In conclusion, kindly state clearly the transition phase which is crucial for the control of the disease. - Information has been added.
- Kindly improve the Figure captions (Figures 3, 5, 6 and 7). Figure captions have been modified.
- Kindly check the broken link (Lines 108-109) - Have been corrected and revised accordingly.
- There are still many old references (more than 5 years and above). - The references that make up the systematic review range from 1980 to 2022. As it is a review that aims to demonstrate the state of the art of the molecular biology of the witches' broom disease caused by perniciosa, it is interesting that references that contributed to the first elucidations are present, even though technological and scientific advances have provided more robust answers on the subject, keeping these references is a way of telling the story. In addition, the review was based on the Prisma guidelines, which guide the use of search strings and inclusion and exclusion criteria, which these studies met, regardless of the year of publication. Therefore, removing these references just because they are old is to bias the review, since criteria for year of publication have not been established.
Language review - The entire document was checked by an English native speaker.
Round 2
Reviewer 2 Report
The authors have made some revisions and improvements on the manuscript. There are several points that need to be revised and clarified, namely:
- Simplify all figure captions. Place the acronyms properly.
- It is recommended not to make the abbreviation "CaOx" for calcium oxalate to avoid readers misunderstanding about its molecular formula (lines 341 and 688).
- Write the molecular formula of hydrogen peroxide correctly (line 420).
- Overall, many studies have found similar responses among different hosts, such as high starch content and high contents of glucose, fructose and sucrose in the apoplast ... (lines 431-432)
Simplify this sentence.
- During the disease process, a decrease in chlorophyll a and b levels is observed shortly after infection, consequently reducing photosynthetic rates (lines 447-448).
How long of this period?
-...that M. perniciosa can grow and develop in the biotrophic phase, employing different plant carbon sources, mainly sucrose, fructose, glucose and glycerol. ... On the other hand, the host uses nonspecific mechanisms, such as an increase in alkaloids, phenolic compounds and tannins, to try to eliminate the fungus, ... (lines 671-681).
Is there a specific mechanism for alcohol compounds occurring molecularly in M. perniciosa and its host?
- According to your main questions, please highlight the following points in your conclusion, such as:
a) Which are the genes related to resistance (or susceptibility) in the interaction between M. perniciosa and its hosts?
b) Which genes are involved in the pathogenicity (virulence) of the fungus?
c) Which genes and proteins are involved in the molecular battle of M. perniciosa x host interaction?
- Please check and do revision on the following references: 46, 49, 61, 81,115, and 121 respectively.
Author Response
Dear referee,
We sincerely appreciate yours contributions in the review and suggestions regarding the manuscript "State of the art of the molecular biology of the interaction between cocoa and witches' broom disease." (Manuscript ID ijms-2003855)
All suggestions from were entirely accepted and highlighted using the “Track Changes” function in the text.
Please find our comments on reviewers' suggestions below.
Sinceriously yours,
Carlos Priminho Pirovani
- Simplify all figure captions. Place the acronyms properly - Captions for all figures have been simplified. Leaving important information for your understanding. The meanings of the acronyms were inserted.
- It is recommended not to make the abbreviation "CaOx" for calcium oxalate to avoid readers misunderstanding about its molecular formula (lines 341 and 688) - Have been corrected and revised accordingly.
- Write the molecular formula of hydrogen peroxide correctly (line 420) - Have been corrected and revised accordingly.
- Overall, many studies have found similar responses among different hosts, such as high starch content and high contents of glucose, fructose and sucrose in the apoplast ... (lines 431-432) Simplify this sentence - Have been corrected and revised accordingly.
- During the disease process, a decrease in chlorophyll a and b levels is observed shortly after infection, consequently reducing photosynthetic rates (lines 447-448). How long of this period? -Information has been added to the text better explain results.
- ...that M. perniciosa can grow and develop in the biotrophic phase, employing different plant carbon sources, mainly sucrose, fructose, glucose and glycerol. ... On the other hand, the host uses nonspecific mechanisms, such as an increase in alkaloids, phenolic compounds and tannins, to try to eliminate the fungus, ... (lines 671-681).
Is there a specific mechanism for alcohol compounds occurring molecularly in M. perniciosa and its host? - The non-specificity of the modulation of secondary metabolites mechanisms was discussed based on the results found by Scarpari et al. (2005). The modulation of secondary metabolites was reported at the biochemical level in studies by Scarpari et al (2005); reported in Garcia et al. (2007) based on the presence of phenolic compounds in histological sections of infected cacao trees, and at the gene level in the works by Teixeira et al. (2014). However, these studies address the modulation of secondary metabolites as a nonspecific response, because they do not present data that support that this response is exclusive/specific to T. cacao only when infected by M. perniciosa.
Although, the increase in secondary metabolites (alkaloids, phenolic compounds and tannins) has been attributed to the basal defense response of T. cacao observed in green brooms, the precise contribution of each compound in the defense or if these compounds are specific for T. cacao in response to M. Perniciosa is so far uncertain.
In the literature, only three articles studied the biochemical interaction of T. cacao and M. perniciosa. Aneja and Gianfagna (2001) analyzed the induction of caffeine synthesis in young cocoa leaves; Orchard and Hardwick (1988) studied photosynthesis, carbohydrate translocation and metabolism in cocoa seedlings infected by M. pernicosa and Scarpari et al (2005) who analyzed biochemical compounds during the development of WBD. These last considers, based on his data, that the increase in levels of alkaloids, phenolic compounds and tannins is a nonspecific response, suggesting that it remains to be seen how the fungus causes or modulates these apparent responses in the plant and to verify these discoveries and advances in area. To understand this complex interaction, additional biochemical studies are needed, as well as detailed analyzes of gene expression.
Gesteira et al. (2007) who generated the first ESTs library from the interaction of T. cacao and M. perniciosa, also warns about the need to investigate other genes from the ESTs library (without disclosing) that may contribute to the understanding of the fungus phase transition mechanisms biotrophic to necrotrophic and relate them to the biochemical changes that occur in the green broom reported by Scarpari et al. (2005).
Data shown in the commented review paragraph are from Scarpari et al. (2005) and so far no other work has brought us a specific answer regarding the modulation mechanism of secondary metabolites in the T. cacao - M. perniciosa interaction. Therefore, characterizing a specific mechanism of secondary metabolites (alcoholic compounds) is a gap in research with T. cacao and M. perniciosa, since, to point out the specificity of this modulation, studies that analyze the interaction of T. cacao are necessary. with different stressor biological agents, in addition to M. perniciosa, so we will know if it is a specific mechanism or a plant response to any stressor biological agent.
Information has been added to the text better explain discussion.
- According to your main questions, please highlight the following points in your conclusion, such as:
a) Which are the genes related to resistance (or susceptibility) in the interaction between M. perniciosa and its hosts?
b) Which genes are involved in the pathogenicity (virulence) of the fungus?
c) Which genes and proteins are involved in the molecular battle of M. perniciosa x host interaction? The answers to these questions are already discussed in the manuscript and detailed in Supplementary Tables #1, #3 and #4. However, information was added in the conclusion to improve the text and value the results (genes and proteins) in common between the different studies. Specific results at the gene and protein level are shown in supplementary materials and discussed throughout the manuscript. Due to the nature of a Systematic Review, we cannot infer more than what is presented on previously published data.
- Please check and do revision on the following references: 46, 49, 61, 81,115, and 121 respectively - References have been corrected and standardized accordingly.